# OpenReview forum: "A Framework for PromptOps in GenAI Application Development Lifecycle"
_ICLR.cc/2024/Conference — Submitted to ICLR 2024_

### Official Review · Reviewer_turR · 2023-10-13

**Soundness:** 1 poor
**Presentation:** 2 fair
**Contribution:** 1 poor
**Rating:** 3
**Confidence:** 3

**Summary:**

This paper presents GenFlow, an innovative framework designed to optimize and simplify the development of applications driven by LLM. Serving as a Prompt as a Service product, GenFlow offers substantial benefits to both developers and non-engineers tasked with managing multiple prompts.

In GenFlow, prompts are abstracted into a highly configurable format, greatly simplifying the management of multiple prompts. At the core of the GenFlow framework lies the GeNode, a foundational component that is both user-friendly and customizable, catering to individual use cases. With the aid of GeNode, users can effortlessly create new prompts, eliminating the need for advanced coding skills, while concurrently ensuring robust version control. This approach revolutionizes the way prompts are handled, offering a streamlined and accessible solution for a diverse range of users.

**Strengths:**

Thank you for your interest in the ICLR. The paper is well-crafted, logically structured, and written in an accessible manner. The exploration of PromptOps is an interesting and noteworthy topic, particularly in the context of the increasing complexity and attention garnered by LLM.

One commendable aspect of this paper is its real-world application and the insights it shares regarding the practical experiences gained. The proposal of the Prompt as a Service concept for efficient prompt management is innovative and thought-provoking.

The choice of prompt in a specific use case as a fundamental component appears well-aligned with contemporary software development needs. This paper offers valuable perspectives on addressing challenges related to advanced language models, making it a meaningful addition to the discourse in this area.

**Weaknesses:**

This paper introduces the concept of GenFlow, aimed at enhancing application development driven by LLM. However, a notable limitation of the paper is the absence of detailed insights into the design and detailed metrics of this service. Consequently, it remains unclear how much improvement GenFlow can offer in comparison to existing methods.

In Section 4, the paper briefly acknowledges the existence of similar services, but it does not delve into what sets GenFlow apart or the innovative aspects that distinguish it from its counterparts.

While Figure 1 provides an overview of the prompt management workflow, the paper primarily focuses on the prompt structure design and configuration settings. An area that warrants further exploration is how GenFlow enhances prompt testing and deployment. A more comprehensive examination of these aspects would offer a more balanced perspective on the tool's capabilities.

**Questions:**

GenFlow presents a promising concept for improving prompt development, and I'm curious about a few aspects:

1. Have you developed a prototype of GenFlow, and if so, could you provide insights into its performance and usability in real-world scenarios?

2. Is GenFlow trained based on a specific language model like ChatGPT, or is it compatible with various language models? Can it be adapted to support models trained by users themselves?

3. In what ways does GenFlow streamline and simplify the prompt testing process? Are there specific features or mechanisms that make this part of the workflow more efficient?

4. Does GenFlow include any provisions or strategies for enhancing prompt deployments, ensuring a smooth and effective transition from development to real-world implementation?

---

### Official Review · Reviewer_uFs3 · 2023-10-29

**Soundness:** 3 good
**Presentation:** 2 fair
**Contribution:** 2 fair
**Rating:** 3
**Confidence:** 3

**Summary:**

The paper describes a tool for easily creating prompts (prompt
engineering) for LLMs, called GenFlow. They integrate a prompt
management which can be used by non-experts.

Prompts are defined in terms of parameter-value pairs. Can become an
API, different prompts can be combined and has a web interface.

**Strengths:**

- An application to create prompts

**Weaknesses:**

- An application tool build with well-known techniques
- It is not clear how easy is to build prompts by non-experts, which
questions the usefulness of the tool
- The are no tests showing the functionality of the system and how it
  compares with manual prompting

**Questions:**

Even with this tool, the used still needs to carefully define the
GeNode to generate the prompts.

It is not clear if the tools could be used to generate more advanced
prompts, like Chain of Thought, self-consistent COT, tree of thought,
etc.

It is not clear once a GeNode is specified, how the systems generates
different training prompts.

The paper does not provide any tests to validate their system.

The authors need to pay attention to the references. None of them are
complete (there is no information of where they were published). Also
reference for PromptSource (Bach et al, 2022) is not included.

---

### Official Review · Reviewer_MEeF · 2023-11-01

**Soundness:** 1 poor
**Presentation:** 1 poor
**Contribution:** 2 fair
**Rating:** 1
**Confidence:** 5

**Summary:**

The paper proposes a framework for prompt operations, similar to what devops operations offer for code and other artifacts.

**Strengths:**

Prompt operations are a vital part of optimizing LLM-based applications and providing solutions that are customized to the nature of prompts as part of the input and ecosystem is important for developer productivity.

**Weaknesses:**

W1- The paper misses an evaluation or usability study that shows how useful the framework is to end users.

W2- The paper does not justify precisely what parts of the current MLOps and DevOps processes are to be rethinked because of the new nature of LLM-based applications in the generative space, that are sensitive to prompt optimization. For example, why isnt code version control not sufficient for prompts?

W3- The paper does not describe architectural details of the framework.

**Questions:**

- Do the authors consider this submission as a contribution to the open source frameworks for LLMs? If so this needs to be clarified in the paper.

- For future submissions, the paper would benefit by showing a case study to how someone could use the framework in practice and from a user study that can demonstrate and evaluate usability.

---

### Official Review · Reviewer_dxyD · 2023-11-15

**Soundness:** 3 good
**Presentation:** 3 good
**Contribution:** 1 poor
**Rating:** 3
**Confidence:** 5

**Summary:**

The paper introduces a novel methodology called “PromptOps” for integrating prompt management into the development lifecycle of Generative Artificial Intelligence (GenAI) systems. It emphasizes the role of prompts in GenAI, proposing a framework that aims to enhance prompt efficiency, reduce bias, and lower development costs. They highlight integration of PromptOps into standard software development practices like CI/CD pipelines, workflows, and APIs. The concept of "Prompt as a Service" (PaaS) is introduced, extending prompt utility beyond development teams to various stakeholders.

The key contributions are 1) Incorporation of Prompt Management into DevOps, 2) GenFlow Tool: This tool democratizes prompt usage by providing an accessible interface for various stakeholders, including non-engineers, to create, modify, and optimize prompts, and 3) Prompt as a Service (PaaS): The paper extends the application of prompts beyond development teams, enabling a broader range of stakeholders to use prompts as integral components in application building.

**Strengths:**

The paper is well-written and easy to understand. The implemented framework provides a practical illustration of the proposed methodology.

**Weaknesses:**

The idea presented in the paper is innovative and holds promise for the future of GenAI application development. However, the lack of results, either from real-world applications or theoretical analysis, is a major limitation. To strengthen the paper's scientific merit, it is essential to include results that demonstrate the framework's effectiveness and potential impact on GenAI development. As it stands, the paper presents a nice idea but falls short of the standards expected for a scientific publication at ICLR.

**Questions:**

No questions.

---

### Meta-Review · Area_Chair_djBW · 2023-12-23

**Metareview:**

While the paper is well-written and presents the idea innovatively, it faces significant criticism for its lack of real-world application results or theoretical analysis to demonstrate its effectiveness. Reviewers note the absence of detailed insights into the design and metrics of the proposed GenFlow tool, and the lack of evaluation or usability studies that show the framework's practical utility. These shortcomings unfortunately hinder its acceptance.

**Justification For Why Not Higher Score:**

The significance of the proposed idea is not verified well.

**Justification For Why Not Lower Score:**

It has a clear presentation and interesting ideas.

---

### Decision · Program_Chairs · 2024-01-16

Reject